# Cryo plus Ultrasound Therapy, a Novel Rehabilitative Approach for Football Players with Acute Lateral Ankle Injury Sprain: A Pilot Randomized Controlled Trial

**DOI:** 10.3390/sports11090180

**Published:** 2023-09-09

**Authors:** Antonio Ammendolia, Alessandro de Sire, Lorenzo Lippi, Valerio Ammendolia, Riccardo Spanò, Andrea Reggiani, Marco Invernizzi, Nicola Marotta

**Affiliations:** 1Physical and Rehabilitative Medicine Unit, Department of Medical and Surgical Sciences, University of Catanzaro “Magna Graecia”, 88100 Catanzaro, Italy; ammendolia@unicz.it (A.A.); valerioammendolia95@gmail.com (V.A.); riccardo.span@gmail.com (R.S.); 2Research Center on Musculoskeletal Health, MusculoSkeletalHealth@UMG, University of Catanzaro “Magna Graecia”, 88100 Catanzaro, Italy; nicola.marotta@unicz.it; 3Physical and Rehabilitative Medicine, Department of Health Sciences, University of Eastern Piedmont “A. Avogadro”, 28100 Novara, Italy; lorenzolippi.mt@gmail.com (L.L.); marco.invernizzi@med.uniupo.it (M.I.); 4Integrated Activities Research and Innovation Department (DAIRI), Translational Medicine, Hospital SS. Antonio Biagio e Cesare Arrigo, 15121 Alessandria, Italy; 5Physical and Rehabilitative Medicine, Casa di Cura La Madonnina, 20122 Milan, Italy; andrea.reggiani1@gmail.com; 6Physical and Rehabilitative Medicine, Department of Experimental and Clinical Medicine, University of Catanzaro “Magna Graecia”, 88100 Catanzaro, Italy

**Keywords:** cryo-ultrasound, cryotherapy, ultrasound, ankle, sports, football, rehabilitation

## Abstract

Background: Acute lateral ankle sprains are common injuries among athletes, but the optimal treatment strategies in elite athletes are still debated. This proof-of-concept study aimed to assess the impact of cryo-ultrasound therapy on the short-term recovery of football players with acute lateral ankle sprains. Methods: Semi-professional football players with grade I or II lateral ankle sprains were randomly assigned to the experimental group (receiving cryo-ultrasound therapy combined with conventional physical therapy) or control group (sham cryo-ultrasound therapy combined with conventional physical therapy). Pain intensity and physical functioning were assessed by the Numeric Rating Scale (NRS) and Foot and Ankle Disability Index (FADI) at baseline (T0) at the end of treatment (T1), after one month (T2), and two months after treatment (T3). Results: After the study intervention, significant between groups differences were reported in terms of pain relief (NRS: 4.08 ± 1.29 vs. 5.87 ± 1.19; *p* = 0.003) and physical function (FADI: 50.9 ± 10.3 vs. 38.3 ± 11.5; *p* = 0.021). However, no significant between group differences were reported at T2 and T3. No adverse effects were reported. Conclusions: Cryo-ultrasound therapy combined with conventional physical therapy can accelerate recovery and early return to sport in elite football players with acute lateral ankle sprains. While this study contributes valuable insights into the potential benefits of cryo-ultrasound therapy, further investigations with a longer follow-up are needed to validate and optimize the application of physical agent modalities in the management of ankle injuries.

## 1. Introduction

Ankle injuries are highly prevalent among professional and amateur sports, with the most typical mechanism of injury involving the combination of plantar flexion with foot inversion [1]. This injury usually shows lateral ligaments of the ankle impairments, with an incomplete tear of one or more ligaments, which could be treated conservatively; indeed, after acute ankle sprains, initial immobilization using a soft splint resulted in faster recovery than simple tubular bandage compression [2].

In this scenario, a short-term immobilization with functional physiotherapy is preferable to 2/3 weeks conventional therapy with plaster [3]. Albeit, there is substantial evidence on the management of ankle injuries [4,5,6,7,8,9], but there is a disagreement regarding the best therapy management for acute ligament injuries in elite athletes, particularly during COVID-19 era [10,11,12,13].

However, nonsurgical therapies can be prescribed for most acute grade I–III lateral ligament sprains with good to excellent outcomes; in detail, physical therapy and several pharmacological treatments and physical agent modalities can be utilized to enhance pain relief and tissue healing, including: diathermy, laser therapy, ultrasound therapy, and other forms of electrical therapies [14,15,16].

In 2012, van den Bekerom et al. [17] suggested that the possible effects of ultrasound therapy seem to be mostly mild and there is the possibly of partial clinical implication, particularly in the short term of the rehabilitation program after these injuries; however, the evidence was insufficient to define an adequate dosage of ultrasound therapy that would be beneficial. On the other hand, ultrasound therapy has been described as a helpful treatment in relieving pain in sports injuries, acting as an edema regulator, presumably by increasing pain thresholds, collagen flexibility, reducing edema and, consequently, inflammation and joint stiffness [18].

Considering the current evidence, cryotherapy appears to be effective in reducing pain, although compared to other rehabilitative approaches, the effectiveness of cryotherapy is still considered as controversial [19,20]. The real effect of cryotherapy on the most frequently treated acute injuries, such as joint sprains or soft tissue injuries, has not been completely explicated [21]. Furthermore, the poor methodology of the current evidence is of concern, so further research is needed to produce proper guidelines of cryotherapy approach and usage, focusing on the development of modalities, durations, and frequencies of ice treatment for dealing with the injury [22,23,24]. In fact, Kwiecien et al. stated that cryotherapy-induced metabolism decreases in inflammation and tissue damage have been proved in an in vivo muscle injury model; nonetheless, analogous evidence in humans is absent. This lack of evidence is prospective due to the insufficient length of application of conventional cryotherapy approach. The conventional application of cryotherapy must be repeated to address this concern [22].

Newly, literature engagement has been raised on the role of cryo plus ultrasound therapy, a physical agent modality that combines cryotherapy (cold therapy) with therapeutic ultrasound [25]. This treatment has been used for soft tissue injuries and inflammatory conditions, mainly in sports medicine and rehabilitation [26]. In this context, the combination of cold and ultrasound, using a single device, might create a synergistic effect, providing both anti-inflammatory and tissue-repairing benefits [26]. Despite these considerations, there is nevertheless a considerable gap of knowledge on the therapeutic effects of cryo plus ultrasound therapy in patients with acute ankle sprain. Furthermore, to date, no previous trial characterized the role of a specific cryo plus ultrasound therapy by a single device in the conventional rehabilitation of elite athletes with acute ankle sprain.

Therefore, the purpose of this pilot randomized controlled trial was to assess the impact of cryo plus ultrasound therapy in the short-term recovery of football players affected by acute lateral ankle sprain.

## 2. Materials and Methods

### 2.1. Participants

This trial was evaluated and registered by the local ethics committee (Comitato Etico Territoriale Regione Calabria) providing the following code: 115/2022, in respect of the Declaration of Helsinki and following the ethical guidelines of the responsible legislative institute. Athletes were educated about the aim of the pilot trial and provided informed consent to collect clinical information for scientific assessments and purposes. All rights of the enrolled subjects in the present study were protected. All authors and research participants were educated in caring about the privacy of the subjects involved.

Inclusion criteria were: (a) adult male; (b) semi-professional football players; (c) I–II grade lateral ankle sprain injury; (d) no persistent instability phenomena or chronic sprains; (e) acute injury (within 2 weeks from trauma); (f) no evidence of bone edema or skin disorder, once the area of intense pain in motion was delimited, the lack of neurological disturbances was investigated and assessed.

Exclusion criteria were: (a) history of recurrent dislocation of the ankle or hyperlaxity of any joint; (b) severe rheumatic diseases and/or collagen diseases; (c) athletes who have received any form of local physical therapy and NSAIDs within the last 2 months prior to injury; (d) athletes who admit to using steroids; (e) any contraindication and/or limitation to the use of a physical agent modality (implantable electrophysiological devices, active neoplasms).

Then, all the athletes included in this pilot trial were randomly allocated with a 1:1 ratio in an experimental group and control group. Randomization was performed by an author not involved in this step of the process of the study using random blocks.

### 2.2. Intervention

Both groups followed the same rehabilitation program in the first week, while the patients were treated with the experimental or sham intervention during the following 2 weeks. During the first week, the approach consisted of a synthetic splinting system for joint immobilization, Canadian crutches for weight-bearing ease, draining massage performed by a physiotherapist with progressive proprioceptive exercises.

After the first week, participants of the experimental group underwent a combination of cryotherapy and ultrasound therapy treatment. The treatment was performed by an expert physiotherapist with a single Cryosound 1.16 device (ELCAP—Giarre CT, Italy) for both treatment groups [6].This device simultaneously delivers cryotherapy and therapeutic ultrasound with the same applicator, not allowing the patient (the blind component of the study) to recognize which type of therapy he was undergoing. The experimental group was subjected to continuous application of cryo-ultrasound, with a temperature of −2 °C and a power of 1.8 watt/cm^2^, as illustrated in Figure 1.

In the control group, a sham treatment was provided without the administration of ultrasound therapy and with the use of only the perceptible sensation of cold, but not at the therapeutic level of cryotherapy. All patients cannot recognize the dummy therapy because the device looks the same as the active one. A 40 min session was performed for both groups, the first 20 min dedicated to rehabilitation recovery of articular function and proprioceptive exercises and the remaining 20 min for the execution of cryo-ultrasound therapy for the active or sham group.

For conventional physical therapy, stretching exercises were conducted in the early phase with closed-chain ankle motions and unloaded dorsiflexion stretching approaches progressing to standing calf stretch and global joint stretching in open-chain [11]. In parallel, progressive strengthening exercises were performed after pre-injury ROM recovery, starting with isometric exercises in both the frontal and sagittal planes. Next, the player moved to isotonic resistance exercises using weights, bands, or therapist manual resistance for all planes pain-tolerated motions [27]. Finally, in the initial stages, PNF exercises started with intrinsic movement of the foot (extension of the toes with plantar flexion of the ankle/flexion of the fingers with dorsiflexion of the ankle) and trainings implemented on a surface of different consistencies, a plank wedge, or a Bosu [28]. Firstly, the subject should start with a wedge plank in an anteroposterior direction; lastly, with greater pain control, a seated Biomechanical Ankle Platform System (BAPS) was utilized for all planned exercises.

### 2.3. Outcome Measures

Pain intensity was considered as the primary outcome measure with a pain numeric rating scale (NRS); considered by any functional activity or movement of the injured ankle. The NRS is an 11-point numerical score from 0 demonstrating “no discomfort” to 10 expressing the “worst pain ever felt”.

Secondary outcome measures were the Foot and Ankle Disability index (FADI), utilized as a degree of functional limitation related with foot and ankle disorders; involving a 26-item sub-score of daily living and pain; each element has a score from 0 (unable to do) to 4 (no difficulty at all). The total possible score is 104 points and a lower score indicates a higher value of functional limitation. Finally, we evaluated quality of life through EuroQol-5D (EQ-5D) index. All patients underwent clinical follow-up at the end of treatment (T1), after 1 month (T2), and 2 months after the end of treatment (T3)

### 2.4. Statistical Analysis

Statistical analysis was performed using JASP Statistical Package (1.16 Amsterdam, The Netherlands). Data were verified for normal distribution according to Shapiro–Wilk test. Homogeneity of variance analysis was assessed via Leven’s test. Categorical or dichotomous variables were summarized with frequencies. Continuous data were presented with means and standard deviations. Effect sizes were presented through Cohen d (95% Confidence interval), all outcome data were calculated for within group and between group differences from different time points. Effect sizes were interpreted as minor <0.5; adequate between 0.5 and 0.8; and large, >0.8. For each test, statistical analyses were 2-tailed and a *p*-value cut-off set at <0.05 was considered significant. The G-Power statistics module from JASP software was used to ensure the assessment of the appropriate sample size. Assuming an alpha level of 0.05 and 80% power, through an effect size of 0.40, with a repeated measure analysis of variance between group interactions, an appropriate sample size was set at 23. This was enlarged to 26 (13 participants per arm) regarding a potential 10% dropping out assumption, and an equal group distribution of subjects included. This study was evaluated and approved by the local ethics committee (Comitato Etico Territoriale Regione Calabria) providing the following code: 115/2022.

## 3. Results

In total, 25 players who met the trial eligibility and who observed the follow-up were evaluated, as depicted in Table 1. Twelve patients were enrolled in the control group, whereas thirteen participants were included in the experimental group. The mean age of the players enrolled in the pilot trial was 22.8 ± 12.62 years. At T0, the groups did not report any demographic and morphometric differences.

A baseline subject assessment reported that both groups had noticeable intensities of pain, with no significant differences between the groups. Nonetheless, starting from T1 and during the treatment plant, the athletes who were enrolled in the experimental group had a significantly larger pain decrease than the control group; however, at T3, similar results were reported in both groups. In parallel, the FADI results showed comparable levels of progress over time for both study groups. Despite these results, patients who received the active cryo plus ultrasound device management displayed significant enhancement in control subjects at T1, but similar results at T3 (Table 2).

In the light of these paired results, we evaluated the differences between the groups at each time point, as shown in the Table 3.

Moreover, we reported the repeated measures analysis as a cumulative evaluation also at the follow-up (for further details, see Figure 2).

## 4. Discussion

This pilot randomized controlled trial aimed to evaluate the short- and long-term effects of cryo plus ultrasound therapy, using a single device, with conventional physical therapy versus conventional physical therapy alone in football players with acute and subacute I–II grade ankle sprain.

At the end of the treatment (T1), active cryo plus ultrasound therapy, in addition to the group treated with conventional physical therapy, allowed the players to obtain high pain relief (NRS, active group: 4.08 ± 1.29 compared to the sham group: 5.87 ± 1.19) and an increase in FADI scale score (active group: 50.9 ± 10.3 vs. sham group: 38.3 ± 11.5); however, similar results were observed at two weeks (T2) and four weeks (T3) of follow-up in both groups, without any side effects.

Conversely, these results legitimize the effectiveness of conventional physiotherapy in the medium term; nonetheless, in the short term, they demonstrated an accelerated recovery and consequent early RTS for the group treated with the synergistic use of cryotherapy and ultrasound therapy.

Unfortunately, there are common mythoi and fallacies in ankle sprain management; these embrace numerous and unnecessary imaging, inapt non-weight bearing, unwarranted immobilization, delayed functional recovery, and inadequate rehabilitative approaches. The application of an evidence-based tailored program that embraces the individual characteristics of the sportsperson could be useful and should be recommended [11].

Logan et al. [29] reported that ultrasound therapy could provide therapeutic effects in the control of pain symptoms especially in sports-related disorders, in edema management, as well as in the reduction in stiffness and functional improvement of joint ROM, plausibly raising the pain threshold, the microstructural flexibility provided by collagen fibers, resolving the edematous framework, as well as the cytokine pattern underlying the inflammation, up to muscle and joint spasms [29]. In 2011, a systematic review on acute ankle sprains concluded that UST was no more effective than a placebo in treating pain and edema, without providing details on the techniques used for measuring UST parameters [30,31]. Doherty et al. [32] suggested that there is a lack of evidence to examine the efficacy of UST in treating acute ankle sprains; the need for rigorous RCTs to demonstrate efficacy has been emphasized. In this scenario, Daniel et al. [33] concluded that the association of UST with taping and PNF training plus tape applications was most advantageous in the treatment and rehabilitation of high ankle sprain injury; indeed, the author suggested that combined effect functional training with UST could be explored by future research.

In fact, Kinkade’s data reported that ice applications and heat packs had matching results [34]; while Costello et al. [35] established that a whole-body cryotherapy (−110 degrees C) administration provided prompt pain relief and, after 15 min, also reduced muscle tone. Lastly, ice could ensure an analgesic result, which might also facilitate therapeutic exercise in early rehabilitative phase [10]. In this scenario, it can be stated that cryotherapy, applied by a specific device, has an immediate and profound analgesic effect on severe nociceptive pain and accelerates the tissue healing process by reflex vasodilatation followed by vasoconstriction [36,37].

Dehghan et al. [38,39] recommended further studies that could measure the combined effects of different rehabilitative approaches including cryo and thermo-therapy, PNF, acupuncture, etc., on the control of pain. Indeed, there is no sufficient evidence that applying ice alone might decrease pain and swelling, as well as enhancing functioning in people with a I–II grade acute ankle sprain [40,41]. Nevertheless, cryotherapy for 3 to 7 days is habitually utilized to decrease pain, diminish swelling and bleeding, reduce the effects of vasoconstriction; furthermore, the administration of spray or ice packs with a 20 min protocol every two hours is commonly considered to be useful [42]. Additionally, it is often suggested that intermittent immersion cold therapy could be supportive for early pain decrease [11]. The innovative element of this pilot randomized controlled trial is instead the usage of a single device for the delivery of the physical agent, without providing an empirical application but guaranteeing the use of an instrumental combination with the same applicator for cryo and UTS [11].

In an injured sportsperson, appropriate timing and safe RTS or competition is the estimated outcome of the rehabilitative approach [43,44]. In this scenario, the sports doctor would finally have to decide on the athlete’s readiness to RTS, following a complex procedure with controversial indication from various bases [45,46,47,48]. On the other hand, getting feedback from rehabilitation team members is imperative, as many of the crucial outcomes might not be assessed or supervised in appropriate settings [49]. The proper RTS timing is often based on the severity of the injury, considering that a common mild ankle sprain might take a 4 week recovery plan and a more deep syndesmotic damage takes 8 weeks for an adequate RTS [50,51]. Since several professional players (particularly in contact sports such as basketball, football and soccer) might often have suffered multiple ankle injuries, with a common joint instability, a safe and rapid approach can be of great help to sports medical personnel [14,52]. Clearly the athlete’s tissue healing response may depend on key factors such as age, genetic patterns, player experience with pre-injury condition, and their following of the rehabilitation protocol [9,53,54,55].

### 4.1. Future Perspectives

Cryotherapy seemed to play an antalgic role in the immediate post-trauma period and also to accelerate recovery; this could suggest that the cryoultrasound approach partially contributes to pain reduction, but larger follow-ups will be needed, even if it will be difficult to objectify them by the acute nature of the disorder [26]. On the other hand, cryotherapy would seem to generate a cooling cone in the tissue through which the high-power ultrasound waves would pass, producing a deep thermal result, which is well tolerated and efficient in reducing painful symptomatology, developing a trophic effect [54], but larger samples with more stringent inclusion criteria will be needed.

### 4.2. Study Limitations

However, this proof-of-concept study is not without its limitations. First, one latent limitation is that the ending follow-up is quite undersized. Nonetheless, because ankle sprains are an acute and often self-limiting disorder, longer follow-up might make it difficult to attribute recovery to intervention alone, compromising both findings and conclusions. Secondly, in this context, it is difficult to be aware of the actual return to play, sport, or tangible pre-injury performance. However, in the athlete’s competitive context, the incompetence to run or jump, and therefore to train, is the most key feature to study, which is why we consider the FADI to be a truly reliable index for this purpose. Third, there is no assessment of a long-term follow-up. Lastly, both cryotherapy and ultrasound therapy, to the best of our knowledge, do not have an appropriate and recommended dosage for these patients, on the other hand this is the first study aiming to analyze the combined approach and to provide, as much as possible, the two physical agent modes using a single device.

## 5. Conclusions

In conclusion, this a pilot randomized controlled trial aimed at evaluating the impact of cryo plus ultrasound therapy, using a single device, on the short-term recovery of football players with acute lateral ankle sprains. Taking together our findings suggested that cryo plus ultrasound therapy can accelerate recovery and an early return to sport in elite athletes. Overall, this study contributes to the understanding of the potential benefits of cryo plus ultrasound therapy in the management of acute lateral ankle sprains of elite athletes. Further studies with longer follow-ups are needed to confirm these positive data and to explore and refine the use of physical agent modalities to optimize the recovery and return to sport of athletes with ankle injuries.

## Figures and Tables

**Figure 1 sports-11-00180-f001:**
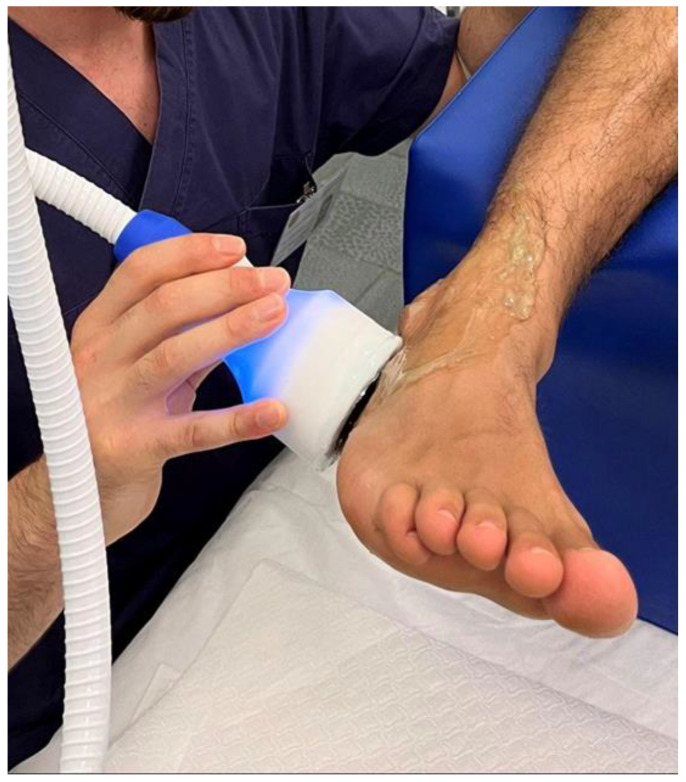
Cryo-ultrasound therapy device.

**Figure 2 sports-11-00180-f002:**
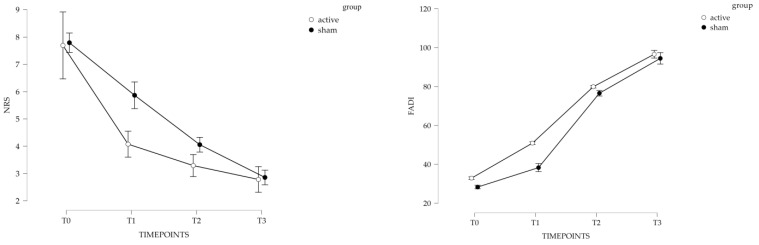
Marginal means plot for NRS and FADI assessment.

**Table 1 sports-11-00180-t001:** Demographic and morphometric characteristics with baseline evaluations of self-reported scales.

Characteristic	Group Exp (n = 13)	Group Cnt (n = 12)	*p*-Value
Age (y), mean ± SD (range)	22.5 ± 12.4 (18 to 41)	23.1 ± 11.5 (21 to 38)	0.114
Weight (kg)	75.1 ± 13 (47 to 88)	77 ± 14 (50 to 92)	0.085
Body mass index, mean ± SD	23.2 ± 4 (19 to 29)	22.9 ± 5 (18 to 30)	0.102
NRS (0–10), mean ± SD	7.69 ± 2.19	7.79 ± 1.19	0.214
EQ-5D-3L Index, mean ± SD	0.5 ± 0.3	0.6 ± 0.2	0.112
FADI (0–104), mean ± SD	32.9 ± 10.5	28.3 ± 10.6	0.079

Abbreviations: Cnt, Control Group; EQ-5D-3L, European Quality of Life 5 Dimensions 3 Level Version; Exp, Experimental group; FADI, Foot and Ankle Disability Index; NRS, Numerical Rating Scale; SD, standard deviation.

**Table 2 sports-11-00180-t002:** Within group differences in the outcome measures for active cryo-ultrasound therapy and control groups.

		T0	T1	∆T0-T1		T2	∆T1-T2		T3	∆T2-T3	
				*p*	ES		*p*	ES		*p*	ES
**NRS (0–10)**	*active*	7.69 ± 2.19	4.08 ± 1.29	0.006	−0.9	3.29 ± 1.05	0.041	−0.3	2.78 ± 0.91	0.083	−0.5
	*sham*	7.79 ± 1.19	5.87 ± 1.19	0.009	−0.6	4.06 ± 1.37	0.052	−0.4	2.86 ± 1.37	0.042	−0.5
**FADI** **(0–104)**	*active*	32.9 ± 10.5	50.9 ± 10.3	0.031	0.7	79.9 ± 8.5	0.005	0.7	96.6 ± 7.6	0.012	0.6
	*sham*	28.3 ± 10.6	38.3 ± 11.5	0.027	0.5	76.6 ± 11.2	0.009	0.5	94.5 ± 7.1	0.039	0.6
**EQ-5D-3L**	*active*	0.5 ± 0.3	0.6 ± 0.2	0.106	0.1	0.7 ± 0.3	0.093	0.0	0.7 ± 0.3	0.0	−0.7
	*sham*	0.6 ± 0.2	0.7 ± 0.3	0.124	0.1	0.7 ± 0.2	0.082	0.0	0.7 ± 0.2	0.0	−0.7

All data are expressed as means ± standard deviations. Abbreviations: ES, Effect size; EQ-5D-3L, European Quality of Life 5 Dimensions 3 Level Version; FADI, Foot and Ankle Disability Index; NRS, Numerical Rating Scale; SD, standard deviation.

**Table 3 sports-11-00180-t003:** Between group differences in the outcome measures for active cryo-ultrasound therapy and control groups.

		T0		T1		T2		T3		ANOVA-RM
		* p * -Value	ES	* p * -Value	ES	* p * -Value	ES	* p * -Value	ES	* p-Value *
** NRS **	* active *	0.744	−0.06	0.003	−0.69	0.212	−0.65	0.534	−0.06	0.002
	* sham *									
** FADI **	* active *	0.2	0.20	0.021	0.57	0.345	0.23	0.386	0.21	0.039
	* sham *									
** EQ-5D-3L Index **	* active *	0.106	0.11	0.127	0.21	0.242	0.13	0.342	0.09	0.128
	* sham *									

Abbreviations: ES, Effect size; EQ-5D-3L, European Quality of Life 5 Dimensions 3 Level Version; FADI, Foot and Ankle Disability Index; NRS, Numerical Rating Scale; SD, standard deviation.

## Data Availability

Data are not available due to ethical restrictions.

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
