# Peer review of "Cryo plus Ultrasound Therapy, a Novel Rehabilitative Approach for Football Players with Acute Lateral Ankle Injury Sprain: A Pilot Randomized Controlled Trial"

_sports, 2023, doi:10.3390/sports11090180_

Round 1

Reviewer 1 Report

This is an interesting study aiming to demonstrate the use of cryo-ultrasound in ankle rehab. While interesting there are several aspects of the manuscript that need improvement.

First, a description of the group assignment is needed. Being that no two injuries are the same, a through description of the group assignment process is so critical to this study.

In that same manner, what controls were in place to the other therapies that each participant experienced. To say that the cyro-ultrasound methods had a impact over the sham all other treatments need to be similar. No mention of those treatment controls are in the methods.

Results: While some of the results (ex FADI) show statistically significant differences, are these clinical significant? Is the change from t0 to t1 truly meaningful in the sham condition? This could be used addressed in discussion but certainly needs more attention.

While I understand that this is a proof or concept study design, the discussion should have a paragraph on future directions based on the findings and ways that this research can be enhanced.

Author Response

This is an interesting study aiming to demonstrate the use of cryo-ultrasound in ankle rehab. While interesting there are several aspects of the manuscript that need improvement.

Thank you for your letter and insightful comments regarding our manuscript entitled “Cryo plus ultrasound therapy, a novel rehabilitative approach for football players with acute lateral ankle injury sprain: a proof-of-concept study”. We would like to express our sincere appreciation for your thoughtful review and valuable comments which help us to further improve this document. Revisions based on your comments are highlighted in the manuscript in yellow and our detailed responses based on each revision are shown below

First, a description of the group assignment is needed. Being that no two injuries are the same, a through description of the group assignment process is so critical to this study.

Thank you for your insightful comment, although we found non-significant differences in the baseline comparison of the two groups, we have examined the inclusion criteria for greater clarity for the reader of the journal.

In that same manner, what controls were in place to the other therapies that each participant experienced. To say that the cyro-ultrasound methods had a impact over the sham all other treatments need to be similar. No mention of those treatment controls are in the methods.

Thanks for the thoughtful comment. Accordingly, we have provided more clarity regarding conventional therapy in methodology in the manuscript as recommended.

Results: While some of the results (ex FADI) show statistically significant differences, are these clinical significant? Is the change from t0 to t1 truly meaningful in the sham condition? This could be used addressed in discussion but certainly needs more attention.

Thanks for the attentive comment. As suggested by the other reviewer, we increased the weight of the comparison with the effect size of each test.

While I understand that this is a proof or concept study design, the discussion should have a paragraph on future directions based on the findings and ways that this research can be enhanced.

Thanks for the helpful comment. We added a paragraph before the study limitations, better clarifying the impact of a future direction of this pilot study.

Reviewer 2 Report

Dear Authors,

I appreciate the opportunity to review your noteworthy proof-of-concept study investigating the effects of cryo-ultrasound therapy on the short-term recovery of male semi-professional football players with acute lateral ankle sprains.

The Manuscript ID “sports-2558988” aligns well with the focus of the Special Issue on “Advances in Lower Extremity Biomechanics and Lower Extremity Injury Risk”. However, after conducting a comprehensive evaluation of the manuscript, I must regrettably convey that, in its current form, I find this research need major revision. While the manuscript exhibits potential in terms of originality and practical application, several significant concerns must be addressed before considering resubmission to this Special Issue of the Sports journal.

An immediate concern pertains to ethical aspects. The omission of the ethical approval code between lines 90 and 93 raises questions. Additionally, the information provided between lines 161 and 163 lacks completeness and contradicts details cited between lines 90 and 93.

Another crucial matter relates to statistical analysis. Despite categorizing your work as a proof-of-concept study, the applied statistical approach should align with that of a randomized clinical trial. Consequently, a more suitable method would involve employing a two-way repeated measures analysis of variance (ANOVA). Such statistical tests can provide valuable insights into the effects of independent variables across multiple measurements, enhancing the comparison and interpretation of groups over time, while also incorporating interaction effects.

Incorporating a mixed model of statistical analysis necessitates the utilization of another measure of practical significance of results. Please incorporate the estimate proposed by Cohen (Psychol Bull. 1992;112(1):155-9. doi: 10.1037//0033-2909.112.1.155), as also outlined in the Lakens manuscript (Front Psychol. 2013;4:863; doi: 10.3389/fpsyg.2013.00863).

Between lines 168 and 170, you state “[…] did not find differences between the groups in terms of age, gender, and location of the sprain […]”. Given that the sample exclusively comprises male players, how was the differentiation between the sexes verified?

Regarding the manuscript's statistical aspects, Table 3 presents p-values from the Friedman test (comparisons over time) and from another test for comparing groups that was not disclosed. Despite Tables 1 and 2 reporting data in mean and standard deviation, a non-parametric test was applied in Table 3. Kindly clarify the rationale behind this choice.

Tables presented within the manuscript necessitate revision. Table 1 displays abbreviations within the table instead of below it. Moreover, “Exp” and “Cnt” appear without full expansion. Table 2 similarly abbreviates variables.

Figure 2 employs fonts distinct from those used in the article's body, featuring low resolution and a smaller font size than the text. Please rectify this.

Review the presentation of p-values in the abstract (lines 29 and 30). The authors employ a colon instead of an equal sign.

References 20, 25, and 27 exhibit inaccuracies. Please rectify these and review all other references for accuracy.

Lastly, the manuscript contains minor typos and spelling errors that might impede readability. Therefore, minor editing of the English language is necessary to ensure a polished and error-free manuscript.

Please understand that the critical points raised in this review are intended to guide you in enhancing the impact and clarity of your manuscript. Addressing these points will undoubtedly unlock the full potential of your research. I anticipate your revisions with great interest.

Minor to moderate editing of the English language is required for this paper.

Author Response

Dear Authors,

I appreciate the opportunity to review your noteworthy proof-of-concept study investigating the effects of cryo-ultrasound therapy on the short-term recovery of male semi-professional football players with acute lateral ankle sprains.

Thank you for your helpful report and insightful comments regarding our manuscript entitled “Cryo plus ultrasound therapy, a novel rehabilitative approach for football players with acute lateral ankle injury sprain: a proof-of-concept study”. We would like to convey our sincere appreciation for your attentive review and valuable comments which support us to further improve this paper. Revisions based on your comments are highlighted in the manuscript in yellow and our detailed responses based on each revision are shown below

The Manuscript ID “sports-2558988” aligns well with the focus of the Special Issue on “Advances in Lower Extremity Biomechanics and Lower Extremity Injury Risk”. However, after conducting a comprehensive evaluation of the manuscript, I must regrettably convey that, in its current form, I find this research need major revision. While the manuscript exhibits potential in terms of originality and practical application, several significant concerns must be addressed before considering resubmission to this Special Issue of the Sports journal.

Thank you for pointing out the potential of our paper, introducing this possible new approach to a widespread disorder. We have subsequently addressed any concerns to improve our manuscript.

An immediate concern pertains to ethical aspects. The omission of the ethical approval code between lines 90 and 93 raises questions. Additionally, the information provided between lines 161 and 163 lacks completeness and contradicts details cited between lines 90 and 93.

Thanks for the thoughtful comment. Unfortunately, there was an error in submitting the manuscript, but we have already emailed the editor and corrected the registered ethics committee code, as recommended.

Another crucial matter relates to statistical analysis. Despite categorizing your work as a proof-of-concept study, the applied statistical approach should align with that of a randomized clinical trial. Consequently, a more suitable method would involve employing a two-way repeated measures analysis of variance (ANOVA). Such statistical tests can provide valuable insights into the effects of independent variables across multiple measurements, enhancing the comparison and interpretation of groups over time, while also incorporating interaction effects. Incorporating a mixed model of statistical analysis necessitates the utilization of another measure of practical significance of results. Please incorporate the estimate proposed by Cohen (Psychol Bull. 1992;112(1):155-9. doi: 10.1037//0033-2909.112.1.155), as also outlined in the Lakens manuscript (Front Psychol. 2013;4:863; doi: 10.3389/fpsyg.2013.00863).

Thank you for your suggestion. We have modified the statistical approach as recommended. Where appropriate we have added effect sizes. According to your suggestion, we defined our paper as a pilot randomized controlled trial.

Between lines 168 and 170, you state “[…] did not find differences between the groups in terms of age, gender, and location of the sprain […]”. Given that the sample exclusively comprises male players, how was the differentiation between the sexes verified?

Thanks for the helpful comment. We have removed the confounding typo, in fact there is no gender comparison in the following table.

Regarding the manuscript's statistical aspects, Table 3 presents p-values from the Friedman test (comparisons over time) and from another test for comparing groups that was not disclosed. Despite Tables 1 and 2 reporting data in mean and standard deviation, a non-parametric test was applied in Table 3. Kindly clarify the rationale behind this choice.

Thanks for the insightful comment. We have completely reformulated the tables, deepened the tests as previously suggested, re-evaluated in the parametric field.

Tables presented within the manuscript necessitate revision. Table 1 displays abbreviations within the table instead of below it. Moreover, “Exp” and “Cnt” appear without full expansion. Table 2 similarly abbreviates variables.

Thanks for the comment. We have corrected the abbreviations of all the tables we provided, as suggested.

Figure 2 employs fonts distinct from those used in the article's body, featuring low resolution and a smaller font size than the text. Please rectify this.

Thanks for the comment, we have reformatted the figures to png, we have also used palatino linotype font

Review the presentation of p-values in the abstract (lines 29 and 30). The authors employ a colon instead of an equal sign.

Thanks for the comment. We have corrected the error in the abstract as suggested.

References 20, 25, and 27 exhibit inaccuracies. Please rectify these and review all other references for accuracy.

Thank you for the careful literature review. We have added the missing DOIs to the references as recommended.

Lastly, the manuscript contains minor typos and spelling errors that might impede readability. Therefore, minor editing of the English language is necessary to ensure a polished and error-free manuscript.

Thanks for the thoughtful comment. We had our manuscript reviewed by an expert native speaker for proper editing.

Please understand that the critical points raised in this review are intended to guide you in enhancing the impact and clarity of your manuscript. Addressing these points will undoubtedly unlock the full potential of your research. I anticipate your revisions with great interest.

Thank you for your further kind comment, we understand the critical points raised, and indeed we are convinced we have been able to provide our manuscript with those facets of insight that were missing.

Round 2

Reviewer 1 Report

Thank you for addressing our concerns.

Author Response

We would like to thank the Reviewer 1 for the comments.

We are glad that our paper was appreciated.

Reviewer 2 Report

Congratulations to the authors for the great work.

On the other hand, the tables still require adjustments in terms of their format and presentation.

Furthermore, the figure is displayed in low quality, which significantly diminishes its readability.

I kindly urge the authors to address these issues to enhance the overall quality of the work for the benefit of the readers.

The manuscript requires minor editing of the English language.

Author Response

We would like to thank the Reviewer 2 for the comments.

We are glad that our paper was appreciated.

According to the suggestions, we revised tables and figures of our manuscript to enhance the overall quality of the work for the benefit of the readers.